# A Novel Approach for Instantaneous Waterline Extraction for Tidal Flats

**Hua Yang, Ming Chen \*** 📧 **, Xiaotao Xi and Yingxi Wang**

College of Information, Shanghai Ocean University, No. 999 Hucheng Ring Road, Shanghai 201306, China; d210900091@st.shou.edu.cn (H.Y.); d190700079@st.shou.edu.cn (X.X.); m210911567@st.shou.edu.cn (Y.W.)
\* Correspondence: mchen@shou.edu.cn

**Abstract:** For many remote sensing applications, the instantaneous waterline on the image is critical boundary information to separate land and water and for other purposes. Accurate waterline extraction from satellite images is a desirable feature in such applications. Due to the complex topography of low tidal flats and their indistinct spatial and spectral characteristics on satellite imagery, the waterline extraction for tidal flats (especially at low tides) from remote sensing images has always been a technically challenging problem. We developed a novel method to extract waterline from satellite images, assuming that the waterline's elevation is level. This paper explores the utilization of bathymetry during waterline extraction and presents a novel approach to tackle the waterline extraction issue, especially for low tidal flats, using remote sensing images at mid/high tide, when most of the tidal flat area is filled with seawater. Repeated optical satellite images are easily accessible in the current days; the proposed approach first generates the bathymetry map using the mid/high-tide satellite image, and then the initial waterline is extracted using traditional methods from the low-tide satellite image; the isobath (depth contour lines of bathymetry), which corresponds to the initial waterline is robustly estimated, and finally an area-based optimization algorithm is proposed and applied to both isobath and initial waterline to obtain the final optimized waterline. A series of experiments using Sentinel-2 multispectral images are conducted on Jibei Island of Penghu Archipelago and Chongming Island to demonstrate this proposed strategy. The results from the proposed approach are compared with the Normalized Difference Water Index (NDWI) and Support Vector Machine (SVM) methods. The results indicate that more accurate waterlines can be extracted using the proposed approach, and it is very suitable for waterline extraction for tidal flats, especially at low tides.

**Keywords:** waterline extraction; bathymetry; tidal flat; remote sensing image





## 1. Introduction

In general, remote sensing applications apply to either land or water analysis; therefore, it is essential to have the land–water masks derived from the same satellite images. Monitoring coastal and wetland regions also needs instantaneous waterline information. For example, the total coastline length of China is about 32,000 km, including 18,000 km of continental coastline and 14,000 km of island coastline. The estimated tidal flats are about 1.5123 million hectares (15,123 km²), accounting for 6.44% of the entire wetland area of China [1]. Waterlines are constantly changing under the influence of natural processes and human activities [2,3]. Therefore, the waterlines are highly dynamic and gradually changing, which makes the information difficult to extract, and not all areas can be surveyed for measurement. How to use modern technologies to improve coastal/island monitoring and obtain accurate coastal/island information is a demanding issue. The remote sensing data provide the instantaneous images taken when the satellites are passing overhead. The instantaneous waterline is the position of the land–water interface at one instant in time [4]. Waterlines are certainly not always horizontal. The wave run-up [5] and wind

setup [6] cause the waterline to change. The instantaneous waterline extracted from the remote sensing image is of great significance for the monitoring of coastal zone changes, ocean dynamics, coastal zone ecological environment protection, resource development, and coastal zone management [7,8].

Traditional waterline monitoring methods mainly rely on field measurements using gauges [4]. These traditional methods are difficult to carry out in areas such as tidal flats and coastal wetlands. The main reason is that this measurement method has a long period, low efficiency, and high cost, and this measurement data are scarce and consume a lot of manpower and material resources in the management and statistics [9,10].

With the development of remote sensing technology and image processing techniques, statistical methods using remote sensing data become an important way to extract waterlines and are successfully applied in many cases [11,12]. Most of the waterline extraction methods are based on multispectral or hyperspectral images. There are three types of waterline extraction methods: edge detection methods; threshold segmentation; and image classification. Edge detection methods detect and create continuous edges (waterlines) on an image; the most common edge detectors include the Canny edge detection [13] and the Sobel edge detection [14]. A well-known threshold segmentation method is the Normalized Difference Water Index (NDWI), which is a band ratio technique that uses Green and SWIR [15] or NIR and SWIR (modified NDWI) [16] bands to generate grayscale images and segment water and land based on thresholds; there is also an automatic thresholding technique based on the histogram—Otsu method [17]. The above two methods are simple and effective; however, they are not effective in tidal flats because there are fewer distinct features that can separate land and water. The third kind of method is based on image classification techniques: machine learning methods such as Random Forest (RF) [18–20], Support Vector Machine (SVM) [21–24], and Logistic Regression (LR) [25,26]. With the recent development of neural networks, there are many methods, such as convolutional neural networks [27] and hierarchical segmentation models [28,29], etc. Image classification methods generally require manual participation and high-resolution images in order to achieve a high recognition rate. Beyond the above-mentioned methods, there is a variety of other methods, such as sub-pixel localization [30,31]. Also, there are various waterline extraction tools, such as CoastSat [32] and CASSIE [3]. The above methods promote the development of waterline extraction; however, in general, it remains a challenging task to extract waterlines in complex topography areas such as tidal flats, and manual digitizing waterlines are frequently required in practical applications.

Multispectral images usually have high spatial resolution and low spectral resolution, and hyperspectral images have low spatial resolution and high spectral resolution. In recent years, with the development of UAV technology, aerial images have appeared and been used for waterline monitoring [33–36]. Aerial images use drones flying at low altitudes, including three-band (red, green, blue) or four-band (red, green, blue, near-infrared), which have high spatial resolution. Therefore, the method of using these images to extract the waterline mainly considers topological features: image shape and texture. Generally, classification methods are used for waterline extraction or photogrammetry techniques. These images have high spatial resolution. The collection is expensive, time-consuming, and requires a large amount of well-trained manpower. It is difficult to achieve a large-scale census. Although commercial satellite images are expensive, ESA provides Sentinel-2 multiband images with 10 m spatial resolution and 11-bit radiometric resolution for free through the internet. Sentinel-2 satellite images are shot periodically and archived. Therefore, there are chances for customers to obtain images suitable for their demands. It is generally used in combination with multispectral and hyperspectral.

Synthetic Aperture Radar (SAR) data are capable of all-weather and full-time [37]. It records information about waterline changes in poor weather conditions. SAR has been well applied in shoreline development [38–40]. However, SAR images have some problems, such as blurred boundaries, low contrast, high grayscale, and susceptibility to noise interference. Sometimes, the contrast between water and land is not strong,

which makes the waterline extraction of SAR images difficult. It needs to be processed simultaneously with other images [41].

Light Detection and Ranging (LiDAR) is point cloud data. The airborne LiDAR is not easily affected by the environment; it can operate in all weather conditions and provide high-precision three-dimensional coordinate data. A digital elevation model (DEM) is typically generated using airborne LiDAR, and then used for separating the land and water [42]. However, since land-based LiDAR cannot penetrate water columns, this results in the inability to accurately extract bathymetry. At the same time, the accuracy of DEM is also affected by spatial resolution and terrain complexity. There is another space-based lidar, ICESat-2, which can reflect water depth. NASA launched ICESat-2 (Ice, Cloud, and Land Elevation Satellite-2) in 2018. The Advanced Topographic Laser Altimeter System (ATLAS), a space-based lidar, was mounted on ICESat-2. ATLAS splits the emitted visible green light (532 nm wavelength) laser into six beams using a diffractive optical element (DOE) and irradiates the beams in pairs of two beams 90 m apart in three rows 3 km apart, centered vertically below the satellite. Thus, ATLAS measures the elevation of the earth's surface in rows of six points. The National Snow and Ice Data Center Distributed Active Archive Center manages ICESat-2 science data.

In tidal flat areas, the spatial and spectral features are affected by various factors such as particle sizes, soil moisture contents, local slopes, sea turbidity, and existing tides. There is no consistent and sufficient correlation between land and water in remote sensing images. In this area, the waterline is blurring, and the uncertainty of extracted waterlines is high. The waterline extraction is the most difficult under the condition of low tides.

Extraction of the waterline in tidal flats from satellite imagery at low tide is difficult. During mid to high tides, most areas of intertidal flats are filled with seawater. So far, none of the studies that have used satellite imagery to extract the waterline have assumed that the waterline is level. This is because the waterline varies with individual waves due to wave run-up on relatively steep beaches. Also, along a shallow coast, the effect of wind setup changes the waterline (https://en.wikipedia.org/wiki/Wind_setup (accessed on 10 December 2023)). However, in relatively calm waters under the weak wind, where the seabed gradient is gentle, it may be possible to extract the waterline from satellite imagery with a coarse spatial resolution (around 10 m or more), assuming that the waterline is level. This paper presents a novel approach to extracting waterlines, especially for tidal flat areas, using bathymetry based on mid–high-tide satellite images. Based on the above-mentioned information, this paper combines Sentinel-2 multiband images and water depth and elevation data obtained by ICESat-2. The former and latter data are used for extracting waterline and for reference water depth data, respectively.

This paper extracts waterlines as follows: (1) It introduces a novel method for waterline extraction based on the assumption that the waterline's elevation is level; (2) It integrates bathymetry and waterline extraction techniques to address the waterline extraction problem: use bathymetry from the high-tide images to improve the waterline extracted from the low-tide images; (3) It transforms the waterline extraction problem into trajectory similarity problem and develops an optimization algorithm to minimizes the area of difference in two trajectories (one trajectory is the isobath from bathymetry and the other is the waterline from the conventional NDWI or SVM method); (4) It derives bathymetry using widely available ICESat-2 or GEBCO data and finally obtains absolute elevation values for the optimized waterline.

## 2. Materials and Methods

### 2.1. Study Area

Two difficult waterline extraction regions were selected as the study areas: Chongming Island (tidal flats) in Shanghai China; and Jibei Island (shallow bedrock and sandy coast) in Taiwan China. Their locations and satellite images are shown in Figure 1.

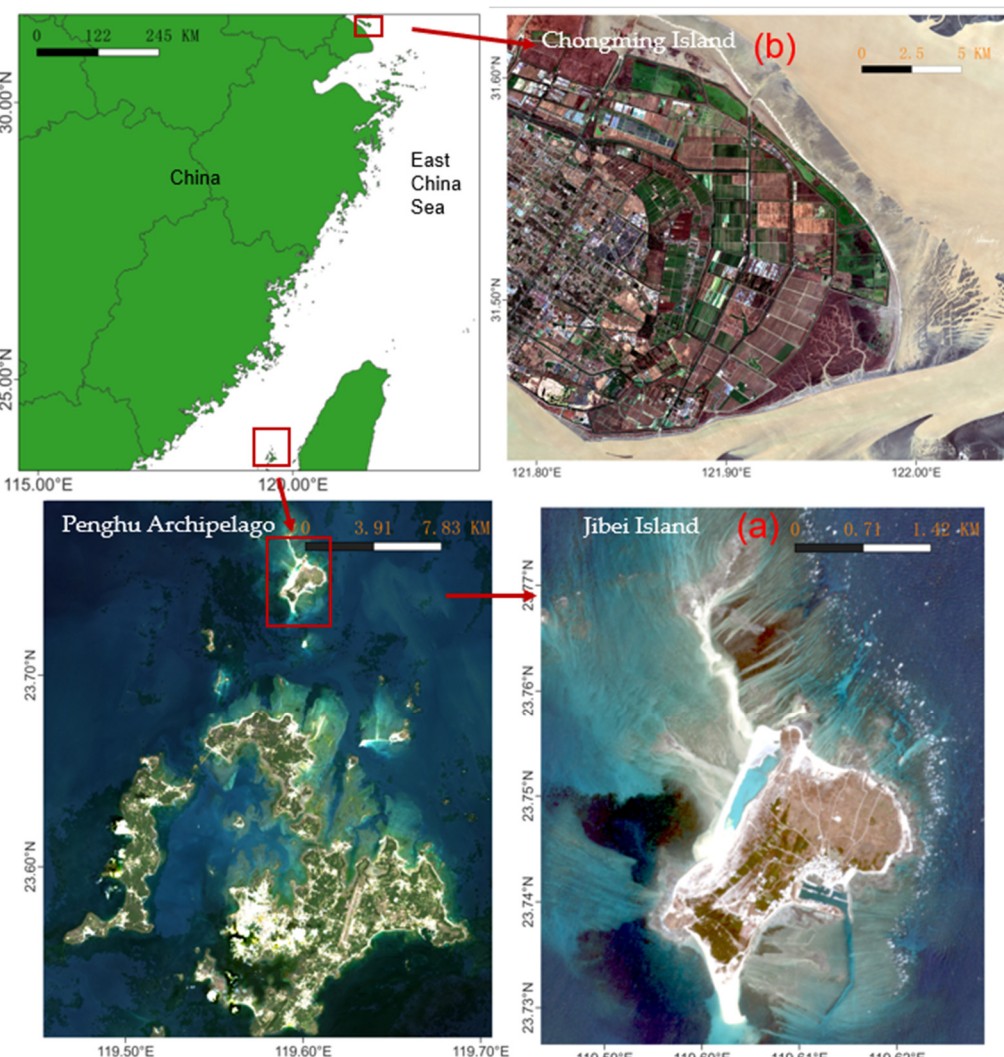

**Figure 1.** Locations of study areas. Bottom row: Jibei Island (**a**) is a part of Penghu Archipelago. Top row, (**b**) is Chongming Island (eastern part). The satellite images (**a**,**b**) are shown using bands 4, 3, and 2 of Sentinel-2 images as red, green, and blue channels.

Chongming Island is located at the mouth of the Yangtze River, 121°50′–122°05′ east longitude, 31°25′–31°38′ north latitude, and is at the east of Chongming Island in Figure 1b. Chongming Island is a typical muddy plain. Chongming Island is formed by the gradual deposition of sediment carried by the runoff of the two channels of Chongming Island South Branch and Chongming Island North Branch and is distributed in a semi-elliptical shape outside the seawall. The Yangtze Estuary is a moderate tidal estuary with a semidiurnal tide and an average tidal range of 2.43 m to 3.80 m. The high concentration of suspended sediment in nearshore water also leads to unclear boundaries between tidal flats and water bodies and is often used as the research object for waterline extraction [43,44].

The second area is located at 23°43′–23°46′N, 119°38′–119°34′E, Jibei Island, the north-ernmost island of Penghu Archipelago, as shown in Figure 1a. Jibei Island is commonly known as "Taiwan Heap". The terrain of the whole island is high in the east and low in the west. The sandy beach and spit are the biggest topographic features of the island. Jibei Island has a vast intertidal zone. The tide situation in the Taiwan Strait is more complicated. Jibei Island has a regular semidiurnal tide; the tidal range is from 1 m to 2.5 m. The island covers an area of about 3.1 km$^2$, with a wide beach and a gentle slope. It is a challenging area for the waterline extraction.

*2.2. Dataset*

Different from the other methods, the proposed novel approach requires high and low-tide satellite images for the same region.

Sentinel-2 is a multispectral imaging mission comprising two satellites, 2A and 2B. Satellite 2A was launched by the European Space Agency (ESA) on 23 June 2015, and 2B was launched on 7 March 2017. The revisit period of one satellite is 10 days, and the revisit period of two satellites is 5 days. The Sentinel-2 satellite carries a multispectral imager (MSI), which can cover 13 spectral bands, and the spectral range covers visible light, near-infrared (NIR), and short-wave infrared (SWIR). The ground resolutions are 10 m, 20 m, and 60 m, respectively. With a width of 290 km, it is used for land monitoring and can provide images of vegetation, soil and water coverage, inland waterways, and coastal areas, and can also be used for emergency rescue services [45]. Only bands 2, 3, 4, and 8 with a spatial resolution of 10 m are used in this study. The captured images' details are shown in Table 1.

**Table 1.** The satellite imagery data for this experiment.

| Study Area | Data Source | Date | Time (UTC) | Tide Status |
|---|---|---|---|---|
| Jibei Island | Sentinel-2 | 20210316 | 022551 | high tide |
| | | 20211111 | 022931 | low tide |
| | ICESat-2 | 20191005 | 142530 | low tide |
| | | 20210402 | 122421 | low tide |
| | | 20211110 | 140101 | low tide |
| Chongming Island | Sentinel-2 | 20230515 | 022531 | high tide |
| | | 20230128 | 023951 | low tide |
| | GEBCO | 20220529 | 121058 | low tide |

General Bathymetric Chart of the Oceans (GEBCO) aims to provide the most authoritative publicly available bathymetry of the world's oceans. It operates under the joint auspices of the International Hydrographic Organization and the Intergovernmental Oceanographic Commission (IOC) (of UNESCO) [24]. This includes global gridded bathymetric data sets, the GEBCO Gazetteer of Undersea Feature Names, the GEBCO world map, Web Map Services, and the IHO-IOC GEBCO Cook Book—a reference manual on how to build bathymetric grids. This paper uses the latest data updated in 2022 as the training data in the Chongming Island experimental area. This is the fourth GEBCO grid developed through the Nippon Foundation-GEBCO Seabed 2030 Project. The grid is used as a "base" Version 2.4 of the SRTM15+ data set, augmented with the gridded bathymetric data sets developed by the four Seabed 2030 Regional Centers.

On 15 September 2018, NASA successfully launched the ICESat-2 satellite. ICESat-2 is equipped with a terrain laser altimeter system (ATLAS) [46], which has been widely used in the elevation measurement of polar ice sheets, sea ice thickness estimation, land elevation measurement, surface vegetation measurement, and other research fields. ICESat-2 has a repeat period of 91 days, with 1387 orbits per period. The ATLAS system is equipped with two lasers, one primary, and one backup; usually, only the primary one is in the working state and emits a single pulse (532 nm) at a repetition rate of 10 kHz, with a pulse width of 1.5 ns, and can obtain overlapping spots with an interval of about 0.7 m along the track and a diameter of about 17 m. The primary laser is split into 6 laser beams, which are arranged in parallel in three groups along the track; each group contains a strong signal and a weak signal, respectively, and the energy ratio between the two is 4:1; the cross-track distance between each group is about 3.3 km, and the cross-track distance within the group is about 90 m. ICESat-2 standard data products are from ATL00 to ATL21. Among them, ATL03 and ATL04 are level 2 products. ATL03 combines photon round-trip time, laser position, and attitude angle data to determine the geodetic position (latitude, longitude, and altitude)

of photon data received by ATLAS. This paper uses ATL03 data; the data are shown in Table 1.

*2.3. Methods*

Unlike current methods, which ignore the waterline's elevation, the proposed waterline approach introduces a new dimension for waterline extraction; this is the general assumption that the waterline's elevation should be level. The waterline's elevation should be level and obtained through bathymetry inversion techniques. In the tidal flats, especially at low tides, the waterlines are very blurring on the low-tide satellite images due to complex spectral characteristics; therefore, it is difficult to derive accurate bathymetry maps using low-tide satellite images. The proposed approach solves this problem by using a mid/high-tide satellite image to derive a reliable bathymetry map first. This solution becomes plausible since the repeatedly acquired satellite images can be easily obtained (for example, the Sentinel-2 series has two satellites, A and B, that can frequently revisit the same region).

The new approach is built on existing waterline extraction and bathymetry extraction techniques. The classic waterline extraction methods NDWI and SVM are chosen as the initial waterline extraction methods of the proposed approach. A simple but efficient bathymetry extraction method—the Stumpf model [47], is selected to derive bathymetry in the proposed approach. The NDWI/SVM method and Stumpf model are only used for illustration and demonstration purposes; it is worth pointing out that the other waterline extracted methods and bathymetry inversion methods can be used in the proposed approach. In the following subsections, NDWI, SVM, and Stumpf models are first briefly described, and then the proposed approach is described.

2.3.1. NDWI

The Normalized Difference Water Index (NDWI) is the water index obtained by McFeeters [15] in 1996 by comparing the spectral differences in different ground features; the equation is as follows.

$$NDWI = \frac{R(\lambda_1) - R(\lambda_2)}{R(\lambda_1) + R(\lambda_2)} \tag{1}$$

where $R(\lambda_1)$ is the reflectivity of the green band, and $R(\lambda_2)$ is the reflectivity of the near-infrared band, which correspond to bands 3 and 8 of Sentinel-2, respectively. This paper uses the common visible and near-infrared bands of Sentinel-2; therefore, it does not use the MNDWI method.

2.3.2. SVM

Waterline extraction is regarded as a binary classification problem. The support vector machine (SVM) method [21–24] is such a technique; it finds a separating hyperplane by maximizing the interval between the target pixel and the background pixel to obtain a decision boundary that satisfies most of the pixels to be classified, making it a classification target. The SVM method can overcome the limitation that the neural network needs a large amount of data for training and is one of the best methods for the classification and regression of small samples.

2.3.3. Stumpf Model

Bathymetry information can be easily derived from the same multispectral images used for waterline extraction. The attenuation degree of the water body's reflectance of blue and green bands can reflect the water's depth. The Stumpf model [47] is a commonly used method, which is based on linear inversion and a logarithmic conversion ratio. The Stumpf model is more stable and robust than the linear model and has high inversion accuracy in areas with clear water and turbid water [48]. This paper is to obtain isobaths through

water depth. It is not necessary to obtain the water depth of each point very accurately. The equation of the Stumpf model is as follows:

$$z = m_1 \frac{\ln(n \times R(\lambda_1))}{\ln(n \times R(\lambda_2))} + m_0 \tag{2}$$

where $n$ is the fixed coefficient of blue and green bands. The blue light and green light have strong penetrating powers to water bodies and can reflect underwater topography. $m_1$ and $m_0$ are empirical parameters as regression coefficients, and $z$ is water depth; $R(\lambda_1)$ and $R(\lambda_2)$ correspond to the reflectivity of the blue and green bands, respectively.

### 2.3.4. The Proposed Approach

Neither NDWI nor SVM waterline extraction methods consider the fact that the waterline's elevation should be level. The relative water depth can be derived from the satellite images using Satellite-Derived Bathymetry. Observing Equations (1) and (2), one can realize that NDWI uses only green and near-infrared bands to separate land and water planimetrically while the Stumpf model uses only green and blue bands to identify the depth of water; it, therefore, makes sense to combine Equations (1) and (2). The proposed approach achieves this. The proposed approach utilizes bathymetry information during waterline extraction, adding extra dimension to tackle the waterline extraction problem. The proposed approach particularly suits difficult waterline extraction cases, such as tidal flats and very shallow water areas: using high-tide images to derive reliable bathymetry and optimize the waterline extracted from low-tide images.

The procedure of the proposed waterline extraction approach is as follows:

1. Prepare high-tide and low-tide satellite images for the targeted area;
2. Obtain the bathymetry map using the high-tide image (use some reference data, such as GEBCO or ICESat-2, to obtain absolute bathymetry; relative bathymetry can still be derived if no reference data are available);
3. Extract the initial waterline from low-tide images using either NDWI, SVM, or other methods;
4. Extract the depth value from the bathymetry map for sample points on the initial waterline and apply robust estimate techniques to estimate the isobath (the contour line, all points on the contour line have the same elevation or depth), which best matches the initial waterline;
5. If the isobath is regarded as the waterline, the process can be ended. Otherwise, the isobath is used to optimize the initial waterline;
6. Form two trajectories: one trajectory is the isobath (from above Step 4), and another trajectory is the initial waterline (Step 3). Apply the proposed area-based optimization algorithm to minimize the area of differences in the above two trajectories. The optimized waterline acts as the final waterline.

The workflow of the proposed approach is illustrated in Figure 2. It is worth pointing out that Step 6 is optional if the isobath is sufficient to be the waterline. The more detailed descriptions of certain steps of the above procedure are given below. It is worth mentioning that suitable DEM along waterline regions can be used instead of bathymetry maps; however, DEM along waterline regions is either inaccurate (out of date) or unavailable (hard to obtain), in the authors' opinion, the bathymetry map derived from the high-tide satellite image is an excellent choice from an efficiency perspective.

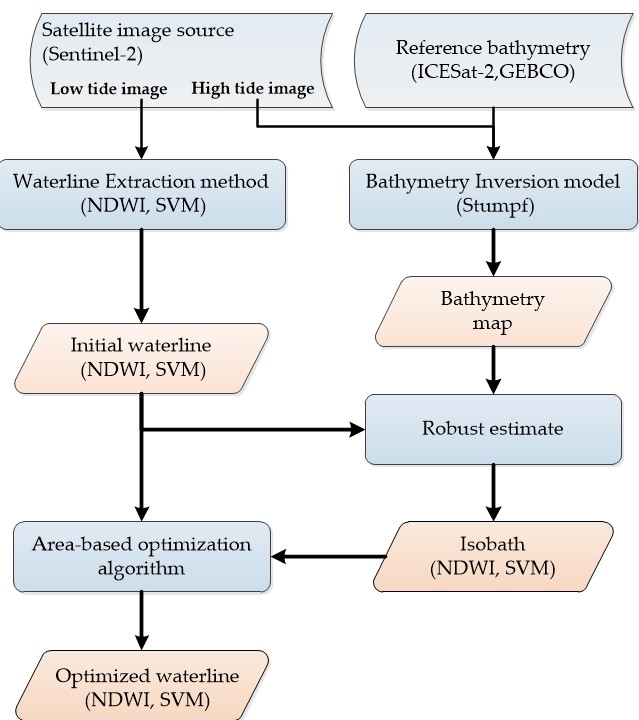

**Figure 2.** Workflow of the proposed waterline extraction approach. The mentioned satellite sensors, waterline extraction methods, and bathymetry inversion models are used in the experiments to demonstrate the proposed approach.

Bathymetry Inversion

Common bathymetry inversion models for Satellite Derived Bathymetry (SDB) are mainly empirical models such as Lyzenga [49] and Stumpf [47]. Recent studies focus on models that use localized optimization techniques such as graphically weighted regression (GWR) [50] or kriging with an external drift (KED) [51] to improve global optimization models. These models usually use limited bands for bathymetry inversion. The log-band ratio method of Stumpf et al. for SDB mapping assumes that the area has a uniform bottom and a log-band ratio of water-leaving reflectance that decreases linearly with water depth. The Stumpf model is employed as the SDB model in the proposed approach for the sake of simplicity.

The Stumpf model requires some reference data in order to inversion bathymetry (see Equation (2)). The GEBCO data are used as reference data to derive the bathymetry of Chongming Island, and ICESat-2 is used as reference data to derive the bathymetry of Jibei Island. The GEBCO data are shown as red dots in Figure 3a. ICESat-2 strips are shown in Figure 3b.

Since ICESat-2 uses a photon counting lidar with high sensitivity, the original photon points cloud data have a lot of noise. Firstly, this paper labels the points to distinguish the water surface and underwater photons. Secondly, after labeling, it is necessary to denoise the labeled data. At present, there are many denoising methods, mainly involving denoising based on raster processing, denoising based on local statistical parameters and denoising based on density spatial clustering (DBSCAN). In this paper, the DBSCAN [52] is used to separate the underwater signal photons. This method is fast and does not need to specify the number of clusters in advance. This method mainly distinguishes signal points and noise points by finding the largest set of density-connected points. The main purpose of this article is to obtain the trend of water depth. The grid points are in the same image, with the same deviation, and do not affect the overall high and low trend. There is no requirement for the accuracy of specific water depth. There is a simple refraction correction; no tidal correction is made.

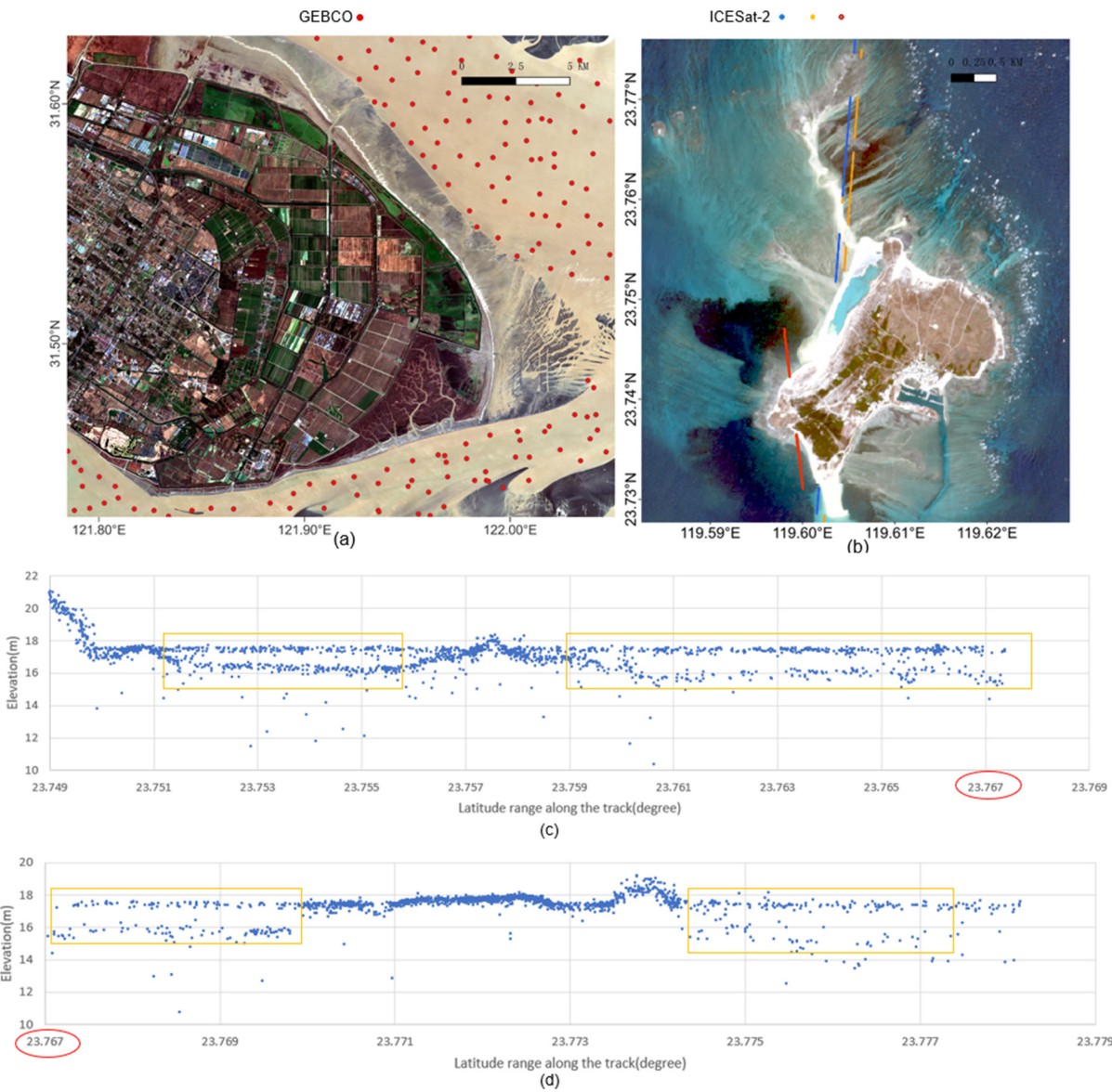

**Figure 3.** Bathymetry reference data. (**a**) GEBCO reference data (red dots) for Chongming Island. (**b**) ICESat-2 strips (blue, red, and orange lines) for Jibei Island. (**c**,**d**) the vertical profiles of ICESat-2 are the yellow strip in (**b**). The data are located in (23°44′52.8″N, 119°36′7.2″E), to (23°46′44.4″N, 119°36′28.8″E). For convenience, the data are truncated from 23°46′1.2″N and divided into upper and lower graphs. The layered photons in the yellow box represent the water body.

Isobath Estimate

The initial waterline is extracted using either NDWI, SVM, or other methods. In order to find the isobath that best matches the initial waterline, samples along the waterline are collected, and samples' corresponding bathymetry values are extracted from the bathymetry map. As a result, each sample has its easting northing coordinates (X,Y) and elevation (Z). In ideal situations, all samples' Z values should be the same; this single Z can easily determine the isobath. Because of the noises/errors both on the waterline and bathymetry map, a fitting is required to estimate a single Z value (Ziso) to determine the best isobath elevation value. Robust estimate techniques can be employed to estimate Ziso. The RANSAC [53] technique is used during the estimate in this study. Using Jibei Island's NDWI initial waterline (generated using a low-tide Sentinel-2 image) as an example, the three pictures in Figure 4 illustrate how its corresponding isobath elevation value is determined on the bathymetry map (derived from a high-tide Sentinel image). In Figure 4a, the black

and red points are the sample points along the waterline (black are inliers, and red are outliers, which are later detected using RANSAC). In Figure 4b, RANSAC is used to find inliers (black points) and outliers (red points) from sample points and obtain the mean elevation of the initial waterline (black line) only using inliers. Once Ziso (black line in Figure 4b) is determined, the isobath is extracted from the bathymetry map, as shown in Figure 4c. In some cases, the isobath can act as the final waterline; however, a further process can be applied to optimize between the isobath and the initial waterline to find the final optimized waterline.

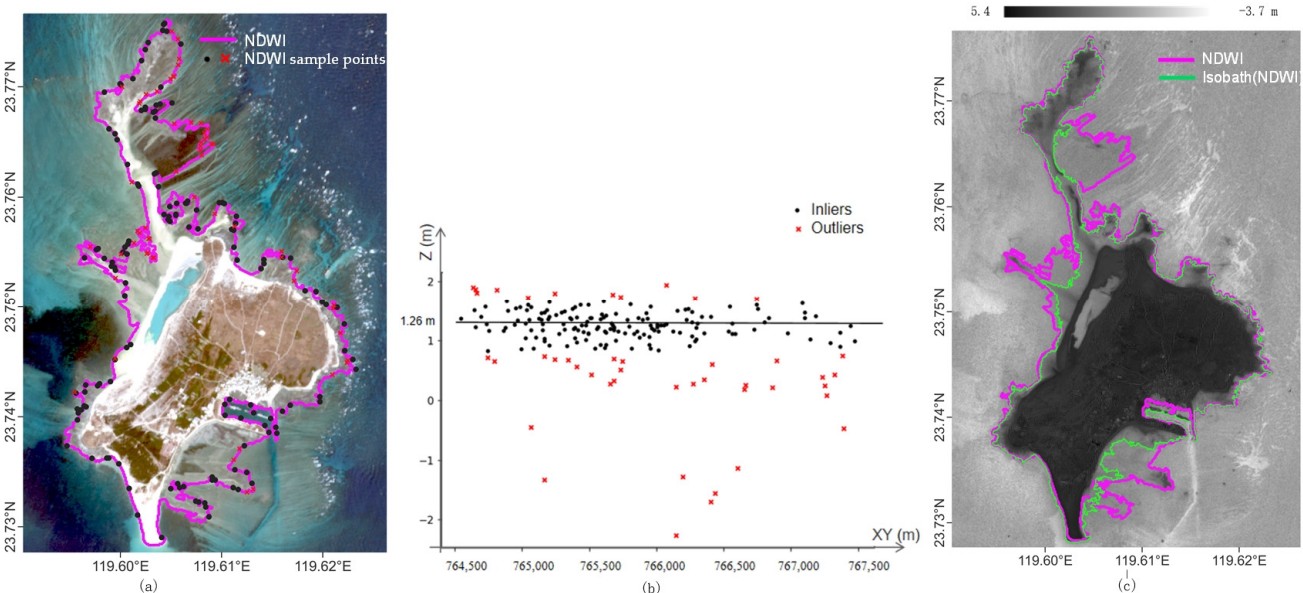

**Figure 4.** Identifying the isobath value of the initial waterline using a robust estimation technique. (**a**) sample points are extracted along the initial waterline (extracted using NDWI in this case). (**b**) Finding the robustly estimated isobath value (1.26 m) using samples from the initial waterline (black points are inliers, and red points are outliers). (**c**) The initial waterline and its corresponding isobath are superimposed on the bathymetry inversion map.

Waterline Optimization

The isobath and the initial waterline can be treated as two trajectories. Pelekis et al. present a method [54], which computes the similarity of two trajectories by the area enclosed by the trajectories. The larger the area, the smaller the similarity. Inspired by this method, it was adapted to find the optimal waterline by reducing the area between the waterline and the isobath. The modified area-based optimization algorithm is shown in Figure 5; the two trajectories have intersections and overlaps. The purple line is the initial waterline. The green line is the estimated isobath from the bathymetry map, and the red line is the corresponding optimized line. The black point $I_i$ is the intersection of the two lines; $Area_i$ is the area enclosed by the two lines. Recorrecting the waterline could be expressed as relocating a line in the $Area_i$ between the intersection points $I_i$ and $I_{i+1}$ of the initial waterline and the isobath; the recorrected line is shown by the red line in Figure 5. The specific steps are as follows: count the intersection points between the two lines first, then count the area enclosed by each intersection according to the intersection points, and calculate the average area as $S_{avg}$. Equation (3) is used to correct the line.

$$
L = \begin{cases} L(I_i, I_{i+1})_{min}, & Area_i \leq S_{avg} \\ L\left(I_i, pmid_1, \ldots, pmid_{j+1}, \ldots, I_{i+1}\right)_{S(L_j, pmid_j)}, & Area_i > S_{avg} \end{cases} \tag{3}
$$

$$S\left(L_j, pmid_j\right) = S_{avg} \times L_j / \sum_{j=1}^{N} L_j \qquad (4)$$

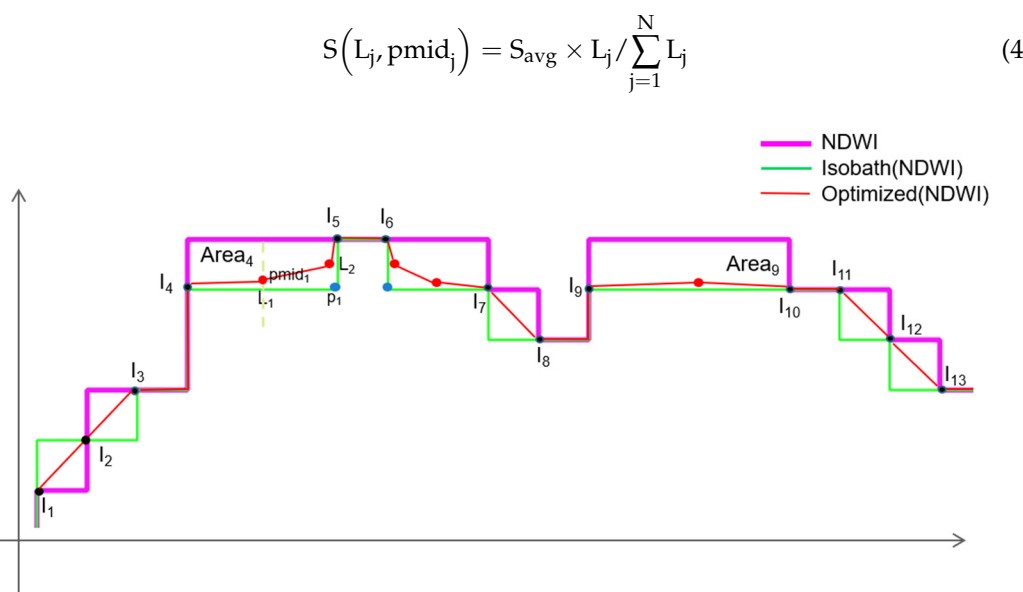

**Figure 5.** The proposed area-based optimization algorithm. The algorithm is to optimize the initial waterline using the estimated isobath and locate a sideline within the two-line siege areas. The green line is the isobath, the purple line is the initial waterline extracted using NDWI method, and the red line is the final optimized waterline.

If the area of the intersecting part is smaller than or equal to the average area $S_{avg}$, the recorrected line between two points is represented by the connection line of the two points $L(I_i, I_{i+1})_{min}$. If it is greater than the average area $S_{avg}$, repeat the following process:

- Split the isobath line into line segments; split the average area according to the number of line segments. The intersection isobath line could be split into N line segments according to the turning point of the raster (blue point $p_j$). For example, $Area_4$ can be split into 2 segments, $L_1$ and $L_2$; each line segment is $L_j$. According to the proportion of each segment to the total length, the average area $S_{avg}$ is divided into N parts, as shown in Equation (4);
- Find a point $pmid_j$ on the vertical line of the line segment $L_j$, such as the red point in Figure 5; this point is in the intersection area of the two lines, and the area enclosed by this point and the line segment $L_j$ is $S\left(L_j, pmid_j\right)$;
- The curve formed by connecting all the relocation points $pmid_j$ is the recorrected waterline.

*2.4. Experimental Design*

Using the datasets described above, two experiments were conducted in Jibei Island and Chongming Island, respectively. High-tide Sentinel images were used to derive bathymetry maps. GEBCO data are used as reference data to derive the bathymetry of Chongming Island, and ICESat-2 is used as reference data to derive the bathymetry of Jibei Island. Low-tide Sentinel images are used to extract the initial waterlines using NDWI and SVM methods. Using the initial waterlines and the bathymetry maps, the isobaths are obtained. The proposed area-based optimization algorithm is used to optimize both the isobath and initial waterline to obtain the final waterline. To quantitatively compare those different results, the reference waterlines in those two experimental areas are established to act as the ground truth. The reference waterlines were manually digitized using the low-tide Sentinel-2 images and closely checked using very high-resolution Google Earth images. Although great care was taken, some errors may still exist in the reference waterlines; therefore, one should be aware that there are some uncertainties in the evaluated accuracies. The initial waterlines extracted using either NDWI or SVM, isobaths, and final water lines are considered the waterline products, which can be compared against reference waterlines.

*2.5. Evaluation*

The labeled waterline is used as the reference waterline. The mean error (Mean) and standard deviation (STD) between the waterline extracted by different methods and the reference waterline are used as the criteria to evaluate the extraction accuracy [30]. The smaller the standard deviation, the more concentrated the data are around its average value. They are calculated as follows:

$$\text{Mean} = \sum_{i=1}^{i=N} \left( P_{i-L} - L' \right)_{min} / N \tag{5}$$

$$STD = \sqrt{\sum_{i=1}^{i=N} \left( P_{i-L} - L' \right)_{min}^{2} / N} \tag{6}$$

where $L'$ represents the reference waterline; $P_{i-L}$ represents any sample point i on the extracted waterline L, $P_{i-L} - L'$ represents the shortest distance from the sample point to the reference waterline, and N represents the total number of samples. According to the geographic coordinates, a sample point is taken pixel-wise. For example, in Jibei Island, more than 7000 sample points are extracted from the waterline.

## 3. Results

*3.1. Jibei Island Waterline Extraction*

For the Jibei Island experiment, two Sentinel-2 multispectral satellite images are used: a high-tide image for bathymetry inversion and a low-tide image for initial waterline extraction. The initial waterline extracted using NDWI or SVM was optimized using its corresponding isobath. For description convenience purposes, the following naming conversions are used in the rest of the text, figures, and tables:

- NDWI: the initial waterline was extracted using the NDWI method;
- SVM: the initial waterline extracted using the SVM method;
- Isobath (NDWI): the corresponding isobath of the initial NDWI waterline from the bathymetry map;
- Isobath (SVM): the corresponding isobath of the initial SVM waterline from the bathymetry map;
- Optimized (NDWI): the final waterline after optimizing the initial NDWI waterline and isobath (NDWI) using the area-based optimization algorithm;
- Optimized (SVM): the final waterline after optimizing the initial NDWI waterline and isobath (SVM) using the area-based optimization algorithm.

All NDWI-related waterlines are shown in Figure 6, and all SVM-related waterlines are shown in Figure 7. Table 2 shows the statistical comparison results between various waterlines and the reference waterline.

**Table 2.** Accuracy comparisons of NDWI, SVM methods, and the proposed approach in Jibei Island.

| Waterline | Mean (m) | STD (m) |
|---|---|---|
| NDWI | 38.35 | 62.24 |
| SVM | 25.23 | 29.66 |
| Isobath (NDWI) | 14.85 | 16.05 |
| Isobath (SVM) | 14.56 | 15.59 |
| Optimized (NDWI) | 15.01 | 16.51 |
| Optimized (SVM) | 14.78 | 16.03 |

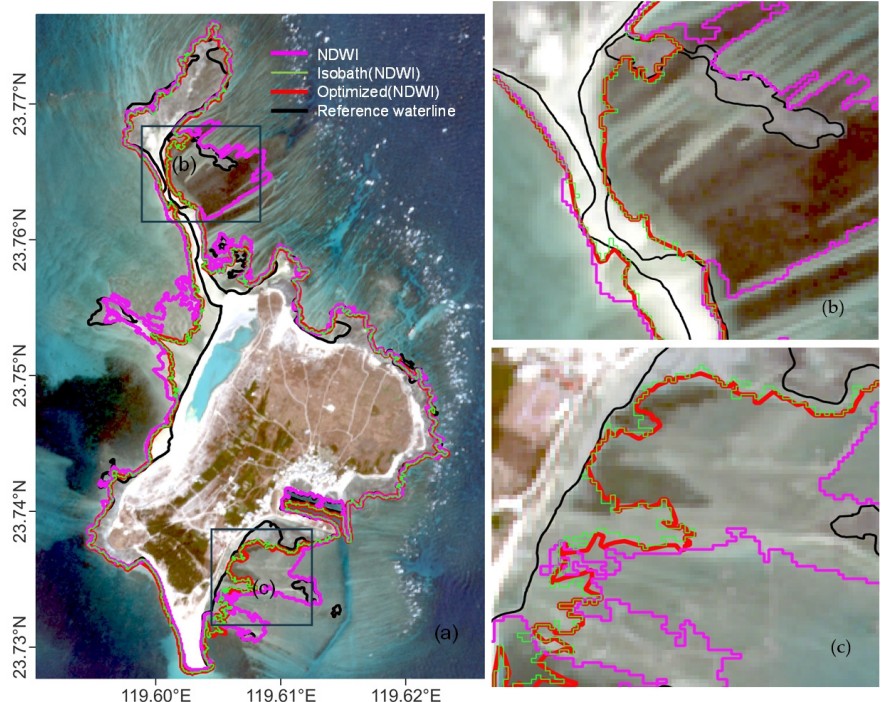

**Figure 6.** Jibei Island's various waterline extraction results based on NDWI method's initial waterline. (**a**) shows the NDWI initial waterline (purple), its corresponding isobath (green), the final optimized waterline (red), and the reference waterline (black). The background image is Sentinel-2 low-tide image captured on 11 November 2021. (**b**,**c**) show two zoom-in regions of (**a**).

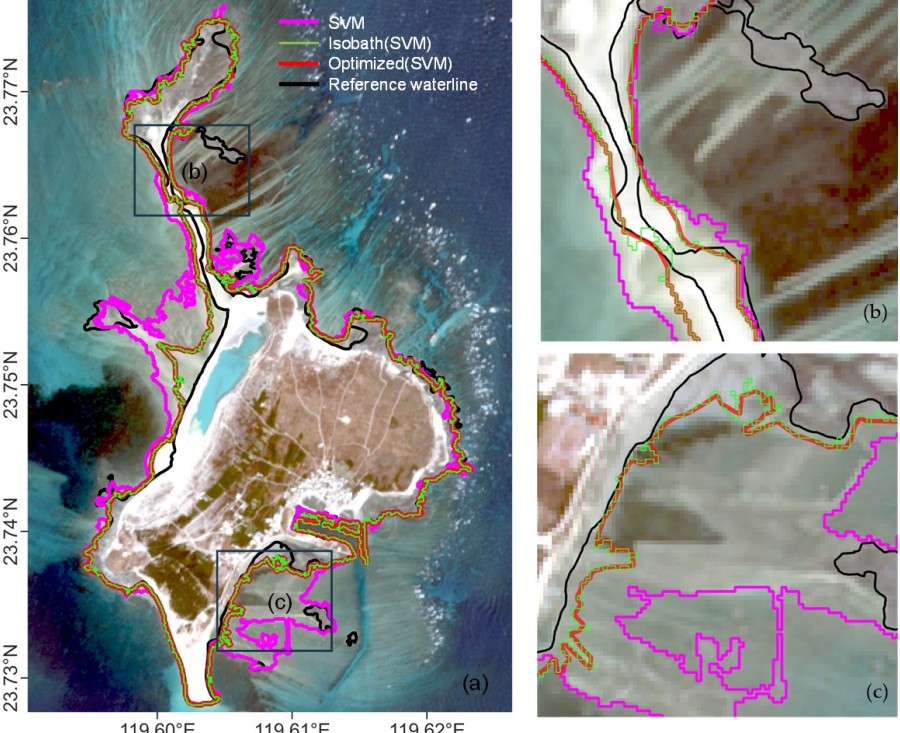

**Figure 7.** Jibei Island's various waterline extraction results based on SVM method's initial waterline. (**a**) shows the SVM initial waterline (purple), its corresponding isobath (green), the final optimized waterline (red), and the reference waterline (black). The background image is Sentinel-2 low-tide image captured on 11 November 2021. (**b**,**c**) show two zoom-in regions of (**a**).

### 3.2. Chongming Island Waterline Extraction

The Chongming Island experiment was conducted in a similar way to Jibei Island: two Sentinel-2 multispectral satellite images were used, a high-tide image for bathymetry inversion and a low-tide image for initial waterline extraction. Isobath was created by the water depth value estimated by the robust estimate method according to the initial waterline. This isobath was used for the optimized initial waterline with the area-based optimization algorithm. The proposed area-based optimization algorithm was used to locate a line, which reduced the area between the initial waterline and the isobath.

All NDWI-related waterlines are shown in Figure 8, and all SVM-related waterlines are shown in Figure 9, respectively. Table 3 shows the statistical comparison results between various waterlines and the reference waterline.

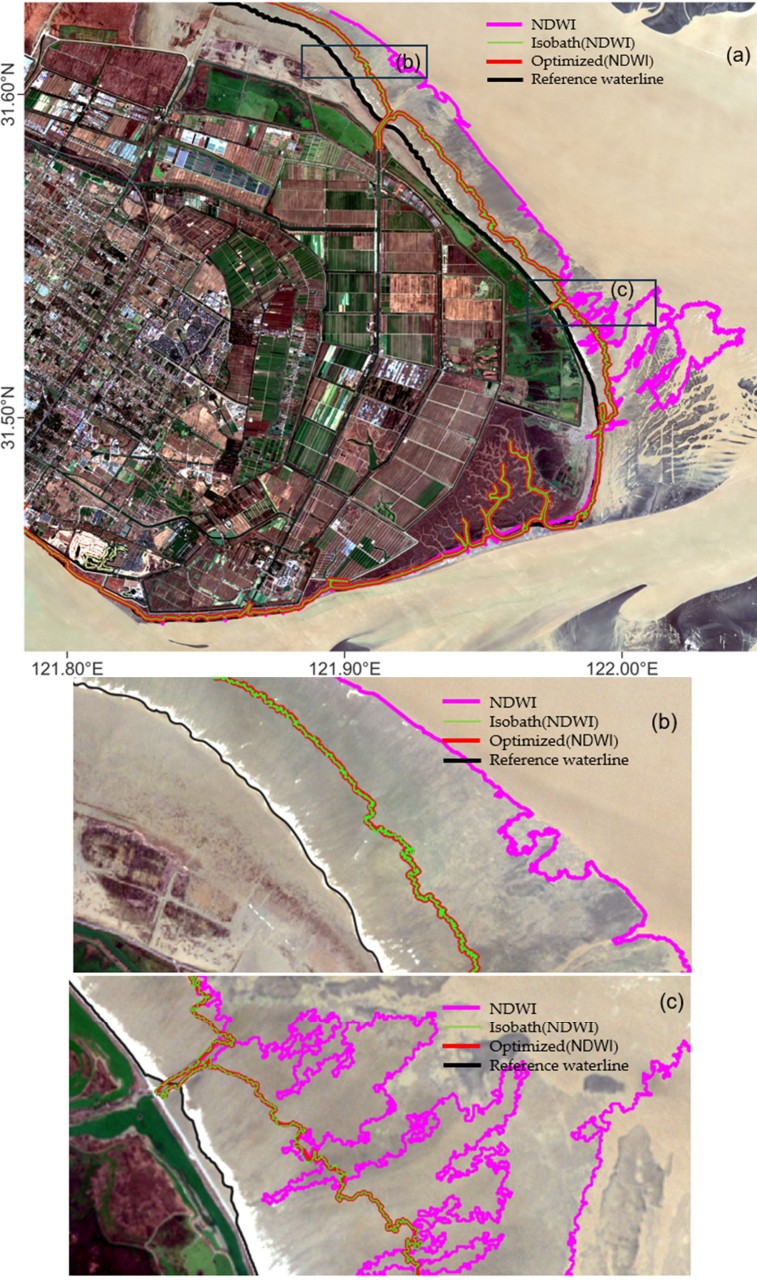

**Figure 8.** Chongming Island's various waterline extraction results based on NDWI method's initial waterline. (**a**) shows the NDWI initial waterline (purple), its corresponding isobath (green), the final optimized waterline (red), and the reference waterline (black). The background image is Sentinel-2 low-tide image captured on 28 January 2023. (**b**,**c**) show two zoom-in regions of (**a**).

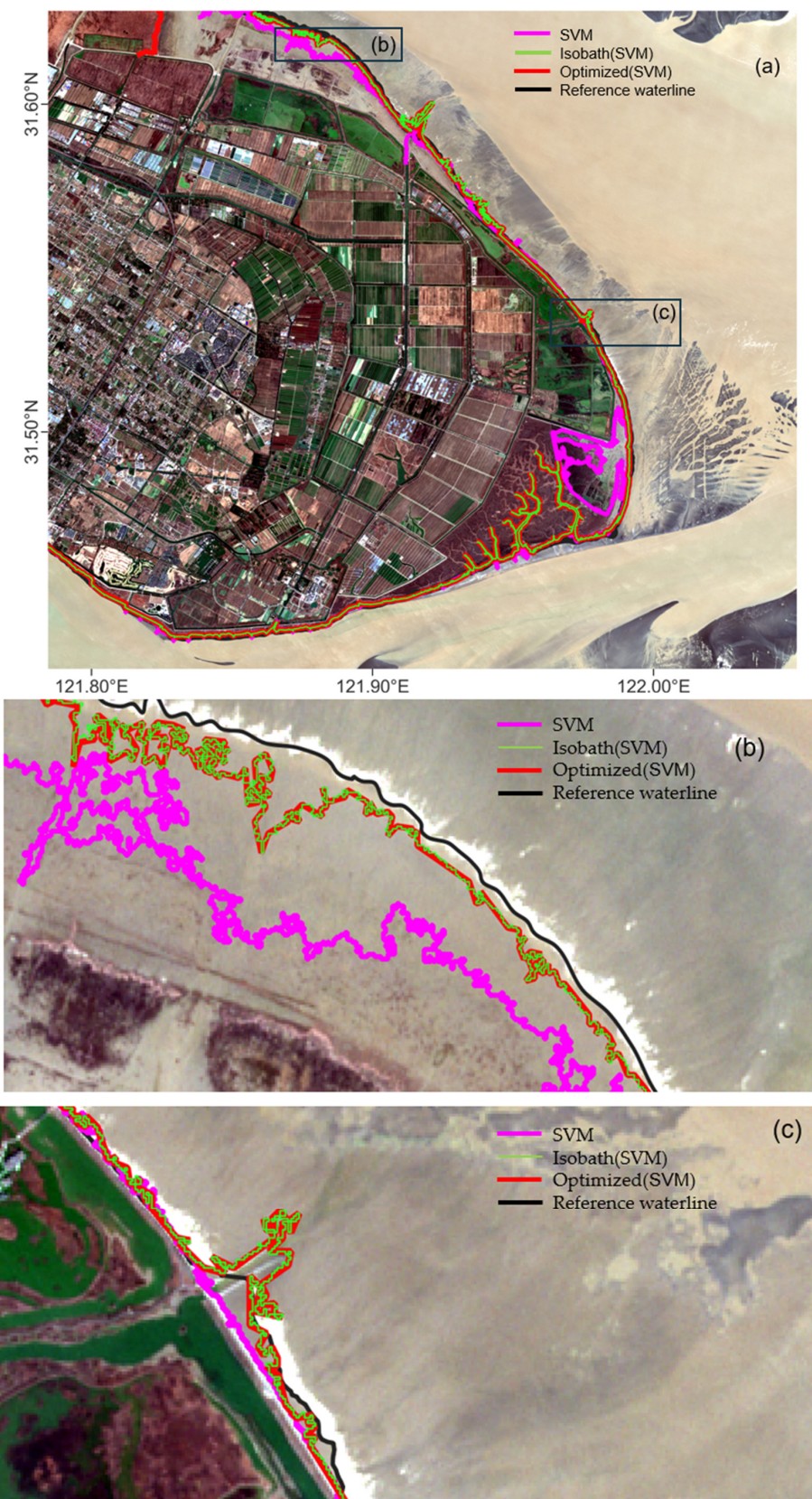

**Figure 9.** Chongming Island's various waterline extraction results based on SVM method's initial waterline. (**a**) shows the SVM initial waterline (purple), its corresponding isobath (green), the final optimized waterline (red), and the reference waterline (black). The background image is Sentinel-2 low-tide image captured on 28 January 2023. (**b**,**c**) show two zoom-in regions of (**a**).

**Table 3.** Accuracy comparisons of NDWI, SVM methods, and the proposed approach in Chongming Island.

| Waterline | Mean (m) | STD (m) |
|---|---|---|
| NDWI | 95.57 | 62.24 |
| SVM | 30.36 | 25.90 |
| Isobath (NDWI) | 26.54 | 20.87 |
| Isobath (SVM) | 19.05 | 17.92 |
| Optimized (NDWI) | 27.34 | 21.15 |
| Optimized (SVM) | 18.93 | 17.51 |

## 4. Discussion

Firstly, it can be seen from Tables 2 and 3 that the waterline extraction method using SVM is better than NDWI. It may be because NDWI is a ratio method; only two bands (band 3 and band 8 of Sentinel-2 images) are used for extraction calculation, while SVM, on the one hand, uses water and land samples for training and considers all bands simultaneously during classification, and the resultant effect is relatively good, especially in shallow water areas. From Figures 6a and 7a, it can be seen that SVM results are much better than NDWI's.

Secondly, from Tables 2 and 3, it can also be seen that the results of isobath (NDWI) and isobath (SVM) are much better than NDWI or SVM's results in both study areas. Further, it is quite obvious that the optimized waterlines, Optimized (NDWI) and Optimized (SVM), are closely related to the isobath waterlines, isobath (NDWI) and isobath (SVM), which proves that the bathymetry information plays a critical role during the waterline extraction process. Although the proposed approach only needs to know the relative water depth, the higher the accuracy of water depth inversion (using some reference bathymetry data such as ICESat-2), the better the extraction of the waterline. In the future, the accuracy of the waterline could be improved by improving the accuracy of water depth inversion.

Thirdly, in the tidal flats, the low-tide satellite images have no obvious spatial and spectral characteristics to separate water and land; it is difficult to derive accurate bathymetry maps. In the proposed approach, the solution is to use a mid/high-tide satellite image to obtain reliable bathymetry maps. The isobaths were generated based on the bathymetry maps derived from mid/high-tide satellite images. The principle is that the isobath is similar to the initial waterline or coincides with an ideal state. This paper uses the isobath line to optimize the initial waterline. It can be seen from Figures 6b,c and 7b,c that the isobaths are close to the reference lines in the low-tide tidal flats. It has the same phenomenon in Chongming, as shown in Figures 8b,c and 9b,c.

This paper uses robust estimate techniques to accurately locate the isobath line. This method assumes that most of the initial waterlines are accurate and obtains some sample points along the initial waterline. There is no deliberate distinction in tidal flats. If there are too many sample points in the tidal flat area, the isobath line may be inaccurate. Furthermore, sampling techniques could be applied to find more accurate initial waterline sample points.

It is noticeable that the waterlines generated using the raster images are not smooth. Straight-line segments are convenient to calculate for the area-based optimization algorithm. Further smoothness could be applied to the final optimized waterlines to improve the product quality. Another issue the proposed approach has not addressed yet is that when simultaneously processing multi-waterlines, currently, only one waterline is applied each time.

## 5. Conclusions

It is of great significance to accurately extract the waterlines of tide flats, especially at low tides. The tide flats have complex landforms, and the spatial and spectral information of the tide flat is not clearly distinguished, making it difficult to distinguish the waterline. Most waterline methods do not perform well in low-tide flat areas. Through the introduction of

the third dimension (waterline elevation/depth) to the waterline extraction, the proposed approach integrates traditional waterline extraction techniques and bathymetry inversion techniques, greatly improving the waterline extraction results compared to the results using some traditional methods such as NDWI and SVM. A further improvement can be made using the proposed area-based optimization algorithm.

It is of great significance to accurately extract the waterline in low-tide flats. The shoal has complex landforms, and it is difficult to achieve manual measurement and large-scale general surveys. Determining the tidal flat edge is very helpful for the general survey of land use and slope calculation in the coastal zone. The 0 m line of water depth based on remote sensing images during the low-tide period is not accurate. The low-tide reference isobath line based on mid/high-tide images have an important reference value for the study of low-tide water depth.

At present, there are many waterline extraction methods. Although this paper uses NDWI and SVM as examples to generate waterlines and optimize them, this approach is not limited to optimizing these two methods. The approach proposed in this paper can optimize the waterline extracted by any method.

The proposed waterline approach shows promising results in two difficult regions where waterline extraction is challenging for traditional methods. Given the nature of the flexible structure and applicability of this approach, there are lots of potentials that could be explored such as trying other than Sentinel-2 sensors and using other than NDWI or SVM methods to generate initial waterlines. To make this approach more practical, how to optimize complicated waterlines needs to be addressed in the future.

**Author Contributions:** H.Y., conceptualization, investigation, methodology, experiment analysis, writing—original manuscript preparation; M.C., conceptualization, supervision—review and editing; X.X. and Y.W., data curation, validation, figures, table preparation. All authors have read and agreed to the published version of the manuscript.

**Funding:** This study was funded by Shanghai Science and Technology Innovation Action Planning, No. 20dz1203800.

**Data Availability Statement:** The original contributions presented in the study are public evaluation data and free satellite data, further inquiries can be directed to the corresponding author.

**Acknowledgments:** The authors gratefully thank the following organizations for providing the experimental datasets: NASA, ESA (European Space Agency), OHB System-AG, and Maxar Technologies for providing satellite data of ICESat-2, Sentinel-2, and Google Earth images, respectively; and the International Hydrographic Organization and the Intergovernmental Oceanographic Commission (IOC) for GEBCO.

**Conflicts of Interest:** The authors declare no conflicts of interest.

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
