# Peer review of "A Novel Approach for Instantaneous Waterline Extraction for Tidal Flats"

_remotesensing, doi:10.3390/rs16020413_

Round 1

Reviewer 1 Report

Comments and Suggestions for Authors

In this paper, the authors suggest a method of waterline extraction from satellite images. They argue that detecting a shoreline at low tide is problematic, especially on tidal flats. High-tide shoreline is better defined. So, they suggest to use a bathymetry map derived at a high tide to help with determining the low-tide shoreline. They utilize an assumption that the water surface is always perfectly horizontal, and therefore the water depth at high tide along the low-water shoreline should be constant. In other words, the low-water shoreline is an isobath on a bathymetry map corresponding to a higher tidal phase. So, the authors suggest to find the average (in some sense) depth value on the initial low-water shoreline, and then replace this waterline with the isobath corresponding to that depth value. The authors then suggest some questionable algorithm of “optimizing” this isobath, which, as follows from tables 2 and 4, does not improve its agreement with the reference low-water shoreline.  

Lest the “optimization” part, the idea of finding the low-tide shoreline as an isobath on a bathymetry map is sound, and the authors seem to demonstrate that it results in an improved shoreline estimate. English, however, requires some work. 

Comments on the Quality of English Language

English, however, requires some work. 

Imprecise use of words starts with the title, “A novel approach for instantaneous waterline extraction for low tidal flats”. Given that, to the best of my knowledge, there are no “high” tidal flats, the word “low” seems to be superfluous. 

Incomplete sentences. For instance, “With the observation that none of current methods considers the fact that the waterline’s elevation should be level. “ (line 12) is not a full sentence (it would be, if started from “None…”, without “With … that” part). Neither are “A global grid at 15 arc-second intervals.”, nor “ Originally published in June 2022. “, (lines180-181). Nor “Because the relative elevation (height or water 261 depth) information can be derived from the same satellite images used for waterline extraction using Satellite Derived Bathymetry (SDB) techniques.”, (lines 261-263).

Overloaded sentences. For instance, “At the same time, the accuracy of DEM is also affected by spatial resolution and terrain complexity, especially in tidal flat areas, due to sudden changes in terrain, it is easy to cause geometric distortion and large waterline extraction errors” (lines 102-104). Please break it in two sentences, to make it understandable.

Repetitions: “Fortunately, repeated optical satellite images are easily accessible in the current days. For the two study areas, multiple Sentinel- 2 can be accessed with an easy.”  (Lines 159-160)

There are nonsense phrases, though a reader might realize what the authors meant. Some example:

“The sandy beach and spit composed of marine terrain are the biggest topographic features of the island. “ (147-148)

A beach composed of marine terrain? I guess the authors meant that the main features of the island terrain (or topography, but not both) are the sandy beach and the spit.

“Sentinel-2 is a multispectral imaging satellite, divided into two satellites” (line 162). I guess the authors meant that Sentinel-2 is a multispectral imaging mission comprising two satellites.

The ATLAS system is equipped with two lasers, usually only one is in the working state, and emits a single pulse” (lines 189-190)

It sounds like the other laser is usually broken, is it?  

Next, “ATLAS emits a total of 6 laser beams” (line 193) - so, how many beams does ATLAS emit?

“The attenuation degree of the water body's reflectance of blue light and green light with increasing depth are difference.” (line 246; I have no idea what the authors meant.)

“…has high inversion accuracy in areas with shallow clear water and turbid water” (line 249) - so, does it has high inversion accuracy in any water? Or in shallow water only?

Grammatically incorrect:

“This solution becomes plausible due to repeatedly acquired satellite images can be easily obtained (for example, Sentinel-2 series has two satellite A and B, can frequently revisit the same region).” (Lines 212-213)

to integrate equations (1) and (2) (line 226) - do you mean: to combine equations?

“The data is located in 23.748N and 23.779N” (line 320) - do you mean: the data points are located between 23.748N and 23.779N?

‘Data’ is a plural 

Author Response

Comments and Suggestions for Authors

Comment 1: In this paper, the authors suggest a method of waterline extraction from satellite images. They argue that detecting a shoreline at low tide is problematic, especially on tidal flats. High-tide shoreline is better defined. So, they suggest to use a bathymetry map derived at a high tide to help with determining the low-tide shoreline. They utilize an assumption that the water surface is always perfectly horizontal, and therefore the water depth at high tide along the low-water shoreline should be constant. In other words, the low-water shoreline is an isobath on a bathymetry map corresponding to a higher tidal phase. So, the authors suggest to find the average (in some sense) depth value on the initial low-water shoreline, and then replace this waterline with the isobath corresponding to that depth value. The authors then suggest some questionable algorithm of “optimizing” this isobath, which, as follows from tables 2 and 4, does not improve its agreement with the reference low-water shoreline.  

Lest the “optimization” part, the idea of finding the low-tide shoreline as an isobath on a bathymetry map is sound, and the authors seem to demonstrate that it results in an improved shoreline estimate. English, however, requires some work. 

Response: Thanks for your kind suggestion. Yes, the isobath line can optimize the initial waterline, the proposed optimization algorithm has no obvious advantages compared to the isobath line, but the proposed method is significantly better than the initial waterline. At the beginning, we assume that the reference waterline should be between the isobath and the initial waterline. Using the area optimization algorithm, we can further optimize the waterline, as shown in Figure 5. However, in reality, the reference waterline may not necessarily be between the isobath line and the initial waterline, as shown in Figures 6 (c), 7 (c), 8 (b) and (c), and 9 (b). In this case, the optimization algorithm is not particularly good compared to the isobath line, but it is better than the initial waterline. Therefore, the optimization algorithm is optional which was described in Line 291-292, and in the Line 501-506, the effect of optimization algorithm depends on the isobath line.

Comments on the Quality of English Language

Comment 1: Imprecise use of words starts with the title, “A novel approach for instantaneous waterline extraction for low tidal flats”. Given that, to the best of my knowledge, there are no “high” tidal flats, the word “low” seems to be superfluous. 

Response: Thanks for your kind suggestion. It has been revised.

Comment 2: Incomplete sentences. For instance, “With the observation that none of current methods considers the fact that the waterline’s elevation should be level. “(line 12) is not a full sentence (it would be, if started from “None…”, without “With … that” part).

Response: Thanks for your kind suggestion. It has been revised.

Comment 3: Neither are “A global grid at 15 arc-second intervals.”, nor “Originally published in June 2022. “, (lines180-181). Nor “Because the relative elevation (height or water 261 depth) information can be derived from the same satellite images used for waterline extraction using Satellite Derived Bathymetry (SDB) techniques.”, (lines 261-263).

Response: Thanks for your kind suggestion. These have been revised.

Comment 4: Overloaded sentences. For instance, “At the same time, the accuracy of DEM is also affected by spatial resolution and terrain complexity, especially in tidal flat areas, due to sudden changes in terrain, it is easy to cause geometric distortion and large waterline extraction errors” (lines 102-104). Please break it in two sentences, to make it understandable.

Response: Thanks for your kind suggestion. It has been revised.

Comment 5: Repetitions: “Fortunately, repeated optical satellite images are easily accessible in the current days. For the two study areas, multiple Sentinel- 2 can be accessed with an easy.”  (Lines 159-160)

Response: Thanks for your kind suggestion. It has been revised.

Comment 6: There are nonsense phrases, though a reader might realize what the authors meant. Some examples:

“The sandy beach and spit composed of marine terrain are the biggest topographic features of the island.” (147-148)

A beach composed of marine terrain? I guess the authors meant that the main features of the island terrain (or topography, but not both) are the sandy beach and the spit.

Response: Thanks for your kind suggestion. It has been revised.

Comment 7: “Sentinel-2 is a multispectral imaging satellite, divided into two satellites” (line 162). I guess the authors meant that Sentinel-2 is a multispectral imaging mission comprising two satellites.

Response: Thanks for your kind suggestion. It has been revised.

Comment 8: “The ATLAS system is equipped with two lasers, usually only one is in the working state, and emits a single pulse” (lines 189-190)

It sounds like the other laser is usually broken, is it?  

Response: Thank you for pointing this out. This expression is ambiguous. It has been revised. ATLAS carries two lasers, one primary and one backup. This information can be obtained from the official website: Space Lasers | ICESat-2 (nasa.gov).

Next, “ATLAS emits a total of 6 laser beams” (line 193) - so, how many beams does ATLAS emit?

Response: Thank you for pointing this out. The ATLAS system is equipped with two lasers, one primary and one backup, usually only the primary one is in the working state, the primary laser was split into six. 

A Laser Beam’s Path Through NASA's ICESat-2 - NASA

Thanks for your kind suggestion. These have been revised.

Comment 9: “The attenuation degree of the water body's reflectance of blue light and green light with increasing depth are difference.” (Line 246; I have no idea what the authors meant.)

Response: Thank you for pointing this out. From the analysis of spectral characteristics, the reflection of water is mainly in the blue and green band, while other bands are absorbed. The attenuation degree of reflectivity in the blue and green band is related to water depth, therefore the different reflectivity can reflect the different water depth, and the Stumpf model calculates water depth values based on blue-green reflectance.

We have revised the description.

Comment 10: “…has high inversion accuracy in areas with shallow clear water and turbid water” (line 249) - so, does it have high inversion accuracy in any water? Or in shallow water only?

Response: Thank you for pointing this out. Under the same water depth conditions, the clearer the water body, the better the inversion model effect. The Stumpf model is stable and robust whether turbid or clear waters. This conclusion can be referenced in reference [47] (Imagery-Derived Bathymetry in Strait of Johor's Turbid Waters Using Multispectral Images).

This expression is not rigorous, we have revised the description.

Comment 11: Grammatically incorrect:

“This solution becomes plausible due to repeatedly acquired satellite images can be easily obtained (for example, Sentinel-2 series has two satellite A and B, can frequently revisit the same region).” (Lines 212-213)

 Response: Thank you for pointing this out. We have revised.

Comment 12:to integrate equations (1) and (2) (line 226) - do you mean: to combine equations?

Response: Thank you for pointing this out. Yes,but it's not just a combination. In this paper, we use the results of equation (2) to recorrect the results of equation 1. We have revised the description.

Comment 13: “The data is located in 23.748N and 23.779N” (line 320) - do you mean: the data points are located between 23.748N and 23.779N?

Response: Thank you for pointing this out. The ICESat-2 data are strip data. The data distribution is linear from point to point, and each point has longitude, latitude and elevation value. Figure 3. (c) and (d) is the two-dimension vertical profiles. We have revised.

Comment 14: ‘Data’ is a plural 

Response: Thank you for pointing this out. These have been revised.

Reviewer 2 Report

Comments and Suggestions for Authors

I am not a remote sensing specialist, but a coastal engineer. So I have assessed the paper from an engineering point of view.

I fully support the fact that determination of the waterline in tidal flats is important and difficult. And indeed, an important element is that a waterline in general is horizontal.  However, when the waterline is horizontal, and I have access to a more-or-less reliable bathymetry, making a waterline is no problem as long a I know the water level at a given moment. Usually that is not a problem, because tidal information is ample available.  The problem is to obtain a reliable bathymetry.

This paper assumes that bathymetry is known from a bathymetry inversion model. For me it is unclear what added value the suggested method gives above a simple contour plotting program added to a bathymetry map.

However, there is an additional problem, not addressed in this paper. Waterlines are certainly not always horizontal. On relatively steep beaches is it very clear, wave run-up causes that the waterline changes with each individual wave. Therefore in such cases a time averaged waterline is used (averaging over a period of a few minutes). With prototype measurements it is shown that this time-averaged waterline is somewhat higher that the water level observed by tidal gauges due to wave set-up. This wave set-up is a function of wave period, wave-height and slope of the beach. So, it can vary along a coast (for a simple explanation see https://en.wikipedia.org/wiki/Wave_setup). On a tidal flat the same phenomenon occurs, but the main actor is the (very) long wave (swell).  

Also along at a shallow coast, the effect of wind-setup is relevant (https://en.wikipedia.org/wiki/Wind_setup).

Both these phenomena make that the waterline is not a horizontal line at a given moment. For example along Jibei island with an westerly wind the real waterline is on the west side of the island much higher that on the east side of the island.

For me it is unclear how correlation between the bathymetry from remote sensing and spectral analysis can lead to more reliable prediction of the waterline. I would expect that at Jibei Island during westerly wind the waterline at the west side is higher than the line following from the bathymetry, while on the east side it is lower.

For coastal engineering applications, as well as for biological research insight in difference of the location of the waterline from the “bathymetry”-determined line is relevant, but I don’t see how this problem is solved with your method.

Author Response

Thank you very much for your comments. Please refer to the Word document for specific responses.

Reviewer 3 Report

Comments and Suggestions for Authors

This manuscript presents a novel approach to extract instantaneous waterline extraction by the utilization of bathymetry, and experiments are carried out in Chongming Island and Jibei Island with Sentinel-2 images. However, is the result of SDB in turbid water practicable? As we all know, the depth through which light can penetrate is extremely limited by the turbidity. Therefore, the applicability of this approach remains to be considered. Besides, in order to improve the writing and understanding of the article, I request your attention to the following questions:

1.    The introduction: The manuscript focuses on the waterline extraction from satellite images. Should LiDAR be considered as images? You mention the advantages and disadvantages of passive optics, SAR and LiDAR, but there is no introduction of what data you choose and why.

2.    Figure 1: Figure (a) and (b) are inversely marked and do not correspond to the text, i.e. Line 136 and Line 145. The scale bar is not clear. The text “Jibei” is underlined in figure(a). The top left figure should be marked with the name of the country or ocean.

3.    Line 161: The sentence is not finished.

4.    Table 1: How to figure out whether the tide status of ICESat-2 is high or low tide? Are there tidal data used in the experiments?

5.    Figure 2: Some of the data types and methods, e.g. Landsat and deep learning, are not used in the experiments. It is not recommended to put it in the flowchart if it has not been used. You should be responsible for the data and methods you have used. Besides, what is NWDI?

6.    Lines 333-334: Please explain the reason of no tidal correction. The tidal gap between ICESat-2 and Sentinel images with different acquisition times cannot be ignored.

7.    Figure 4: The longitude and latitude coordinates of figure (a) and (c) are wrong. Unit of the color bar is missing in figure (c), and I think here the color refers to elevation rather than water depth, which should not be described as deep and shallow.

8.    Section 3.2: Is SDB feasible in the water area around Chongming Island? And whether the accuracy of SDB is evaluated. How to avoid introducing the error of SDB into waterline extraction?

Author Response

(The authors gave the same response as above.)

Round 2

Reviewer 2 Report

Comments and Suggestions for Authors

The advantages of this method over simply using a good bathymetric map are still not clear to me.

Author Response

Thank you very much for your comments.

1. The bathymetry map is generated through depth measurement, anchor points positioning, and accuracy correction. To ensure the quality of the bathymetry map, it is necessary to carefully check and analyze the accuracy of the data horizontal control, elevation control, and anchor points positioning, as well as the density of anchor points and the accuracy of contour drawing. Due to tidal and other reasons, multiple measurements are required in the same area. Therefore, obtaining a good bathymetry map is difficult and requires a lot of manpower and material resources. Since this paper does not require accurate water depth, this method can easily and quickly obtain large-scale waterlines.

2. In order to get water depth. Appropriate surveying line span and map scale must be selected. The direction of the surveying line is generally perpendicular to the isobath lines. Therefore, the depth measurement points are sparse, as can be seen from Figure 3(a) which are General Bathymetric Chart of the Oceans (GEBCO) bathymetry data. The water depth generated through bathymetry inversion is continuous, and the trend of isobaths and waterline are more accurate, as can be seen from Figure 4(c).

Therefore, it is difficult to get a good bathymetric map. Due to limitations in the natural environment, not all areas can be surveyed for water depth measurement. This method based on remote sensing is simple and easy to implement.